# Detecting the Early Flowering Stage of Tea Chrysanthemum Using the F-YOLO Model

**Chao Qi, Innocent Nyalala**  **and Kunjie Chen** *

Department of Agricultural Machinery (AM), College of Engineering, Nanjing Agricultural University, Nanjing 210031, China; chaoqi.njau@gmail.com (C.Q.); innocentnyalala@gmail.com (I.N.)
* Correspondence: kunjiechen@njau.edu.cn; Tel.: +86-13951007707

**Abstract:** Detecting the flowering stage of tea chrysanthemum is a key mechanism of the selective chrysanthemum harvesting robot. However, under complex, unstructured scenarios, such as illumination variation, occlusion, and overlapping, detecting tea chrysanthemum at a specific flowering stage is a real challenge. This paper proposes a highly fused, lightweight detection model named the Fusion-YOLO (F-YOLO) model. First, cutout and mosaic input components are equipped, with which the fusion module can better understand the features of the chrysanthemum through slicing. In the backbone component, the Cross-Stage Partial DenseNet (CSPDenseNet) network is used as the main network, and feature fusion modules are added to maximize the gradient flow difference. Next, in the neck component, the Cross-Stage Partial ResNeXt (CSPResNeXt) network is taken as the main network to truncate the redundant gradient flow. Finally, in the head component, the multi-scale fusion network is adopted to aggregate the parameters of two different detection layers from different backbone layers. The results show that the F-YOLO model is superior to state-of-the-art technologies in terms of object detection, that this method can be deployed on a single mobile GPU, and that it will be one of key technologies to build a selective chrysanthemum harvesting robot system in the future.

**Keywords:** tea chrysanthemum; flowing stage detection; deep convolutional neural network; F-YOLO

## 1. Introduction

Numerous studies have shown that tea chrysanthemum can significantly inhibit the activity of carcinogenic substances, and boasts distinct anti-aging, cholagogic, antihypertensive, and other effects at the early flowering stage. At present, tea chrysanthemum is harvested at the early flowering stage, and the harvesting process is labor-intensive and time-consuming. With the development of artificial intelligence, many jobs can be performed by selective harvesting robots [1–3]. The robotic harvesting process is divided into two steps. In the first step, a computer vision system is used to detect tea chrysanthemum at the early flowering stage. In the second step, the manipulator harvests the chrysanthemum, guided by the detection results. In these two steps, detecting chrysanthemum at the early flowering stage is critical. The detection results not only serve as a guide for the manipulator to harvest chrysanthemum in the subsequent operation, but also determine the detection accuracy in chrysanthemum harvesting. Although in recent years, methods based on deep convolutional neural networks (CNNs) have made remarkable achievements in object detection tasks [4–10], under agricultural application scenarios, it is still difficult to build a lightweight network for a selective harvesting robot that can adapt to complex unstructured scenarios.

In this paper, we propose a lightweight CNN called Fusion-YOLO (F-YOLO), which can adapt to illumination variation, occlusion, and overlapping scenarios. Its performance is shown in Figures 1 and 2. A CNN architecture was designed to understand common features of these scenarios, and to take advantage of the synergy between them. In many works, fusing features of different scales is an important means of improving detection

performance and computational speed [11–13]. Low-level features have a high resolution and contain more details, but due to few convolution operations, these features have less semantic information and more noise. In contrast, high-level features have more semantic information, but have a low resolution and poor ability to perceive details [14,15]. Apparently, the key to improving the performance of detection models is to efficiently fuse the features of different convolutional layers.

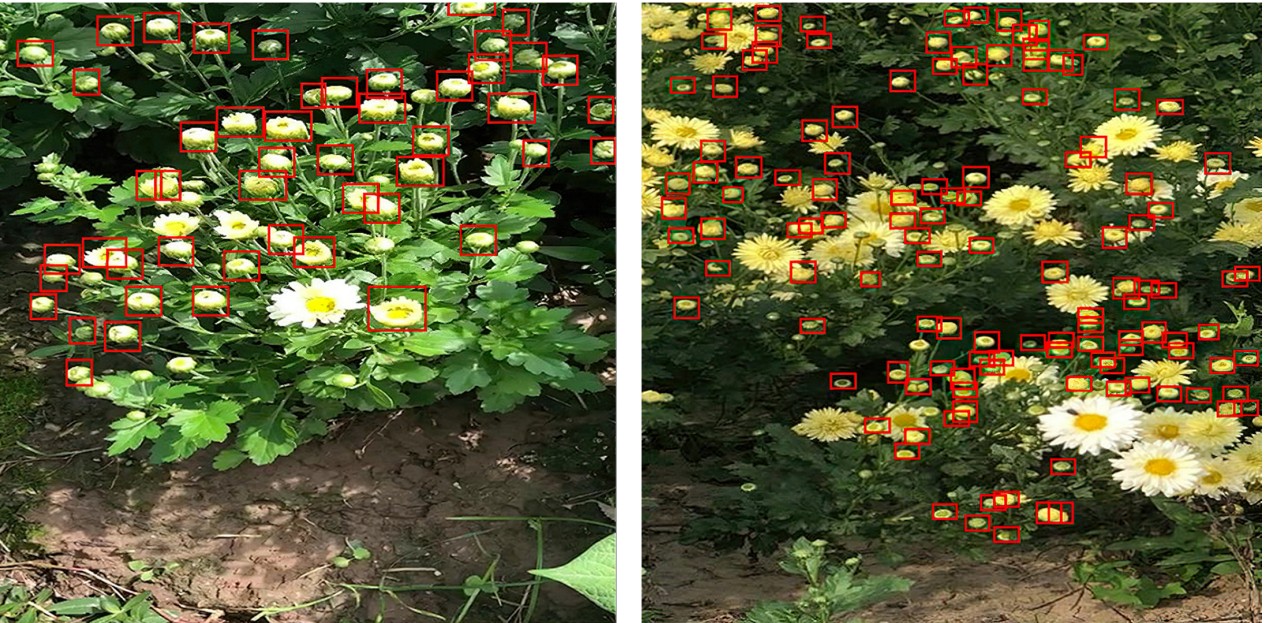

**Figure 1.** The detection performance of the Fusion-YOLO (F-YOLO) model under illumination variation, occlusion, and overlapping.

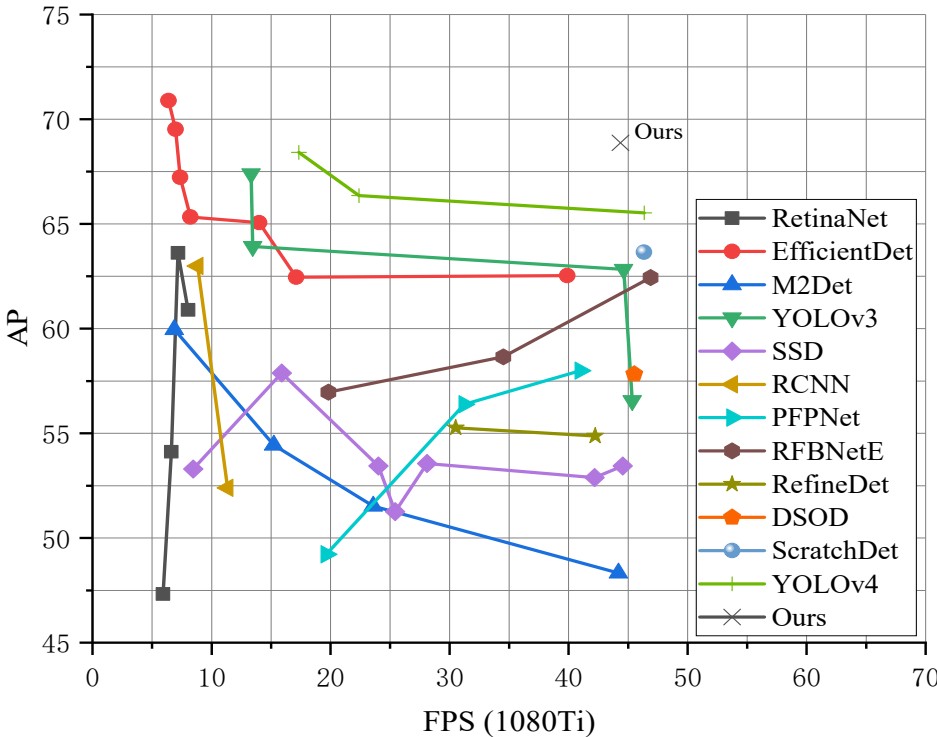

**Figure 2.** The comparison results between the proposed F-YOLO model and 12 state-of-the-art model frameworks.

With the significant improvements in computer performance and the rapid development of deep learning, the advantages of feature fusion have become increasingly prominent. Feature fusion algorithms can be classified into three categories: algorithms based on Bayesian decision theory, algorithms based on sparse representation theory, and algorithms based on deep learning theory. In object detection, algorithms based on deep learning theory are the mainstream—that is, multi-class features from several neural networks are fused to obtain fused features, examples of which include spatial pyramid pooling (SPP) structure [16], pyramid pooling podule (PPM) [17], and atrous spatial pyramid pooling (ASPP) [18]. The information of a large number of small objects can be obtained by feature fusion, thus improving the detection accuracy and speed.

Before the introduction of a feature pyramid network (FPN), most object detection methods, as with classification methods, used single-layer features for processing, and did not add high-level semantic information to the low-level feature map. However, in object detection, high-level semantic information is particularly important. FPN uses multi-scale feature fusion to fuse the high-level and low-level feature maps, which has been adopted by YOLOv3 [19] and other single-stage object detection methods.

Considering that object detection algorithms usually only contain the feature information of the image, instead of semantic information, Zhang et al. [20] proposed detection with enriched semantics (DES) based on the single-shot multi-box detector (SSD) framework, in order to fuse the semantic information of the low-level and high-level features of an image. The segmentation module, which can improve the semantic information of the low-level feature map, and the global activation module, which can improve the semantic information of the high-level feature map, can solve the problem of the low detection efficiency of the SSD for small objects. The FPN algorithm improves the feature extraction ability and detection accuracy of a network by combining high-level semantic information with low-level image information. However, the FPN network architecture is artificially designed, and the fusion effect is not optimal. For this reason, Ghiasi generated a new feature pyramid network called NAS-FPN, using neural architecture search (NAS) technology [21]. Eventually, by combining NAS-FPN with several backbone models in the RetinaNet framework, a high object detection accuracy was realized for current object detection models. The architecture of extended neural networks usually leads to more computation, which makes it impossible for most people to undertake heavily computational tasks, such as object detection. Therefore, a cross-stage partial network (CSPNet) [22] was proposed, which can be implemented on ResNet, ResNeXt, DenseNet, etc. The fusion effect not only reduces the computing cost and memory usage of these networks, but also significantly improves the reasoning speed and accuracy.

Feature fusion aims to transform features into common subspaces in which they can be combined linearly or nonlinearly. The latest development of deep learning shows that a CNN can estimate any complex function [23–26]. Therefore, we built a separate fusion CNN to fuse different function modules. To realize the detection task, a variety of data enhancement methods were fused, and specific loss functions were used to train these modules. By doing so, these function modules could better understand the features of chrysanthemum, and the performance of the lightweight network in complex unstructured environments could be improved in an end-to-end manner.

Feature fusion convolutional neural networks have become increasingly powerful as they have become deeper [27] and more extensive [28]. However, extending the architecture usually requires more computation, and lightweight computation has received more and more attention. The task of harvesting chrysanthemum at a specific maturity stage usually requires shortening the reasoning time on small devices, which poses a serious challenge to computer vision algorithms. Although some methods are specially designed for mobile CPUs [4,29,30], the depth-wise separable convolution techniques adopted by these methods are not compatible with industrial integrated circuit (IC) design, examples of which include application-specific integrated circuits (ASICs) and edge computing systems. In view of this, a lightweight network based on feature fusion is proposed in this paper, which can be

deployed on a mobile GPU [31] without compromising performance. The main purpose of the network design is to enable the architecture to achieve more abundant gradient combinations and reduce the amount of computation. By dividing the feature map at the base layer into two parts, and then merging them through the proposed cross-stage hierarchy, the gradient flow can propagate through different network paths. By switching series and transition steps, a large correlation difference is produced in the propagated gradient information. The contributions of this paper are as follows:

1.  A fusion detection model was designed that alternates between cross-stage partial DenseNet (CSPDenseNet) and cross-stage partial ResNeXt (CSPResNeXt) as the main network, and is equipped with several combination modules. The model can achieve abundant gradient combinations and effectively truncate the redundant gradient flow;
2.  We studied the impact of different data enhancement methods, feature fusion components, dataset sizes, and complex unstructured scenarios on the performance of the F-YOLO model, and proved the superiority of the F-YOLO model by comparing it with a number of state-of-the-art detection models;
3.  A lightweight detection model was designed for detecting chrysanthemum at the early flowering stage, which can adapt to complex unstructured environments (illumination variation, occlusion, overlapping, etc.). Anyone can train an accurate and super-fast object detector using a common mobile GPU, such as the 1080 Ti.

The organization of this paper is as follows. Section 2 explicates the lightweight network F-YOLO that can adapt to complex unstructured environments. Section 3 introduces the setup of the chrysanthemum dataset and the performance of F-YOLO. Finally, Section 4 gives a brief summary of this paper.

## 2. Materials and Methods

In this paper, A lightweight CNN was proposed that can adapt to complex, unstructured environments (illumination variation, occlusion, overlapping, etc.). The network architecture is deep, both vertically and horizontally; that is, it has both top-down connection and horizontal connection, as shown in Figure 3. In this section, we provide a brief overview of the system and then discuss the components in detail.

The proposed algorithm, called F-YOLO, consists of three components: the backbone, the neck, and the head. For the backbone component, the main network is CSP-DenseNet [32]; in addition, a CBL (convolution + batch normalization + leaky ReLU) [33] module and an SPP addon module are added to the component. For the neck component, the feature extraction network is CSPResNeXt [34], the multi-scale fusion network is the FPN+ pyramid attention network (PAN) structure, and a CBL addon module is added to the component. The detection component is a post-processing step, involving an iterative region proposal and non-maximum suppression distance intersection over union (DIOU-NMS) [34], based on the anchor frame, in order to increase the score of chrysanthemum detection and the performance of the detection task. The Mish [35] activation function, drop block [36] regularization method, and the generalized intersection over union (GIOU) [37] loss function are used in the whole network.

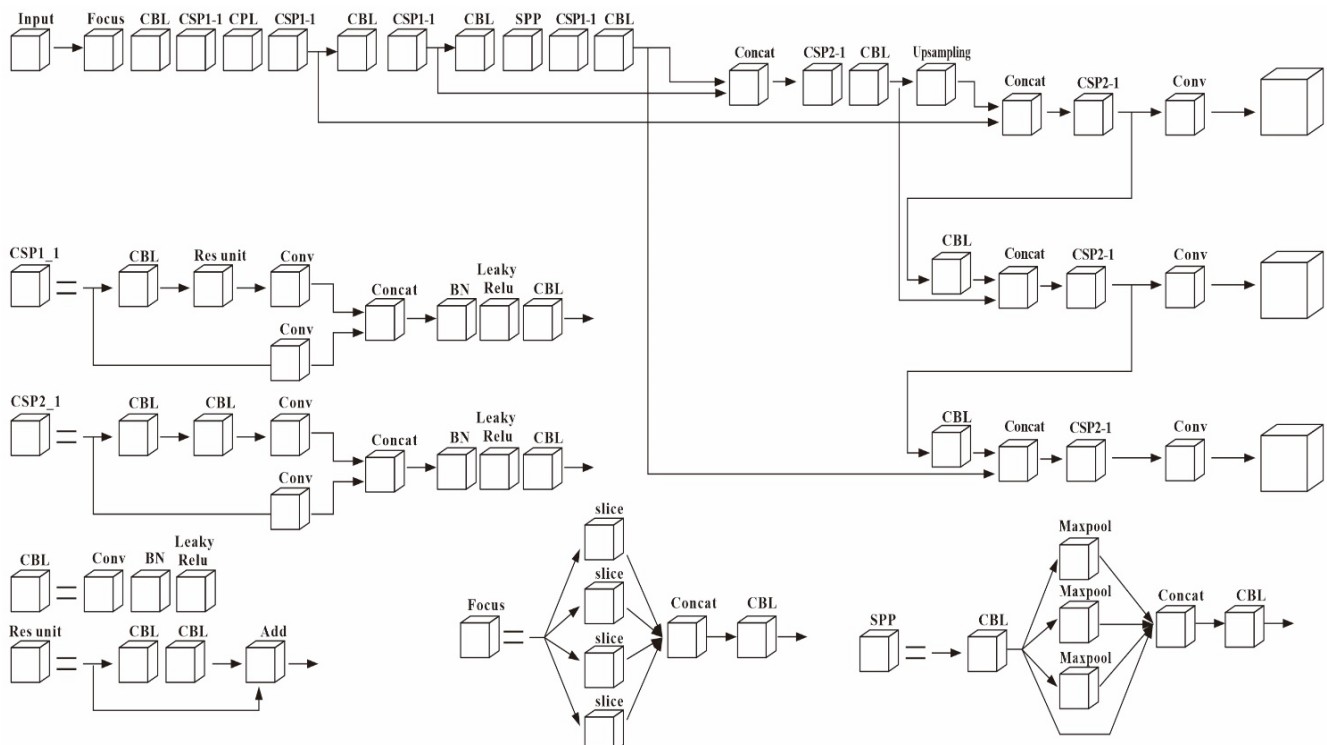

**Figure 3.** Structure of the proposed F-YOLO network. Convolution + Batch normalization + Leaky ReLU (CBL), Cross-Stage Partial (CSP), Spatial Pyramid Pooling (SPP), Batch Normalization (BN).

### 2.1. Experimental Setup

The experiments were carried out on a server with NVIDIA GeForce GTX 1080ti, Conda4.8.3, and cuDNN7.6.5. The CPU was an Intel (R) core (TM) i7-8750h CPU at 2.20 GHz, and the operating system was Ubuntu16.04. The basic detection frameworks were CSPDenseNet and CSPResNeXt. In the training process, the key hyperparameter settings were as follows: momentum = 0.9; gamma = 0.1; weight decay = $5 \times 10^{-4}$; maximum value = 10,000; and primary function = 0.2. In addition, the learning rate decreased to 1/10, 1/100 and 1/1000 at 3000, 6000, and 8000 iterations, respectively, and the optimizer used was a stochastic gradient descent (SGD).

### 2.2. Dataset

The chrysanthemum datasets used in this paper were collected from chrysanthemum breeding bases in Sheyang County in Jiangsu Province, Dongzhi County in Anhui Province, and Nanjing Agricultural University in Jiangsu Province from October 2019 to October 2020. An Apple X mobile phone was used to capture images with a resolution of 2436 × 1125 (458 ppi). All images were taken under natural light, covering several types of interference, including illumination variation, occlusion, and overlapping.

Chrysanthemums were captured in these images at three stages: the budding stage, the early flowering stage, and the full-bloom stage. The budding stage refers to the stage when buds on the main stem or branch of the chrysanthemum plant can be identified by the naked eye and the petals are not opened. The early flowering stage refers to the stage when the petals are not fully opened, and the full bloom stage refers to the stage when the petals are fully opened. A total of 12,040 chrysanthemum image samples were divided into training, validation, and test datasets at the ratio of 6:3:1. The statistical results of the relevant data sets are shown in Table 1.

**Table 1.** Statistics of the datasets used for the construction of the model.

| Label | Original | | | Augmented |
| | Budding Stage | Early Flowering Stage | Full-Bloom Stage | Preprocessed Images |
|---|---|---|---|---|
| Training | 2529 | 2273 | 2422 | 1806 |
| Validation | 422 | 379 | 403 | 301 |
| Test | 1265 | 1137 | 1210 | 903 |
| Total | 4216 | 3789 | 4035 | 3010 |

In routine project training, the average precision (AP) of a small object is generally much lower than that of medium and large objects [38]. Therefore, in image preprocessing, four random images were selected for random scaling, flipping, and mosaicking, and were scaled to 608 × 608 pixels, which greatly enriched the detection dataset. In particular, random scaling increased the number of small objects, making the network more robust. The dataset is shown in Table 1. Each prediction module had three prior anchors of different scales. Through $k$-means clustering [39], nine prior anchors—(21,10), (31,16), (34,41), (45,21), (60,28), (60,78), (76,38), (108,52), and (190,101)—of the chrysanthemum dataset were obtained. At the same time, the data of four images were computed, so that the mini-batch size did not need to be large, and one GPU could achieve good computation results. Cutout operation was performed to simulate occlusion. Preprocessed chrysanthemum images are shown in Figure 4. The pre-processed chrysanthemum image 608 × 608 × 3 was transformed into a 304 × 304 × 12 feature map via the focus slicing operation, and then one convolution operation of 32 convolution kernels was performed to finally form a 304 × 304 × 32 feature map.

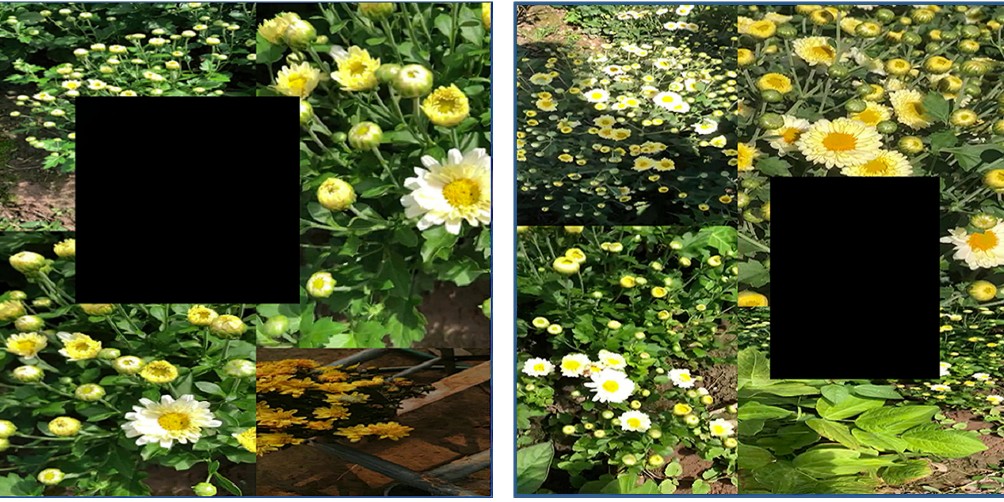

**Figure 4.** Preprocessed chrysanthemum images.

*2.3. Backbone*

2.3.1. CSPDenseNet

The architecture of one stage of the proposed CSPDenseNet network is shown in Figure 3. The stage of the CSPDenseNet network is composed of a partial dense block and a partial transition layer. In the partial dense block, the feature maps of the base layer in the stage are split into two parts through channel: $x_0 = [x_0', x_0'']$. Between $x_0'$ and $x_0''$, the former is directly linked to the end of the stage, and the latter will go through a dense block. All steps involved in the partial transition layer are as follows. First, the output of dense layers, $[x_0', \ldots, x_k]$, will go through a transition layer. Second, the output of this transition layer, $x_T$, will be concatenated with $x_0''$ and go through another transition layer, and then

generate an output $x_U$. The feed-forward pass and weight updating of CSPDenseNet are shown in Equations (1) and (2), respectively.

$$
\begin{aligned}
x_k &= w_k * [x_0, x_1, \ldots, x_{k-1}] \\
x_T &= w_T * [x_0, x_1, \ldots, x_k] \\
x_U &= w_U * [x_0, x_T]
\end{aligned}
\tag{1}
$$

$$
\begin{aligned}
w_k' &= f(w_k, g_0, g_1, g_2, \ldots, g_{k-1}) \\
w_T' &= f(w_T, g_0, g_1, g_2, \ldots, g_k) \\
w_U' &= f(w_U, g_0, g_T)
\end{aligned}
\tag{2}
$$

where, * is the convolution operator; $[x_0, x_1, \ldots]$ is the connection $x_0, x_1, \ldots$, $w_i$; $x_i$ is the weight and output of the $i$-th dense layer; $f$ is the function of weight updating; and $g_i$ is the gradient propagating to the $i$-th dense layer. It was found that a large amount of gradient information is repeatedly used to update the weights of different dense layers, due to which different dense layers may repeatedly learn the copied gradient information.

As we can see, gradients from dense layers are integrated separately. On the other hand, the feature mapping $x_0'$, which does not pass through the dense layer is also integrated separately. For gradient information updated by weights, both sides do not contain repeated gradient information of the other sides. In general, the CSPDenseNet network proposed in this section retains the advantages of DenseNet feature reuse characteristics, but at the same time prevents excessive repeated gradient information by truncating the gradient flow. This idea was realized by designing a hierarchical feature fusion strategy and by applying it to partial transition layers.

### 2.3.2. Partial Dense Block

The purpose of the partial dense block design is to (1) increase gradient paths—the number of gradient paths can be doubled by splitting and merging, and thanks to the cross-stage strategy, the shortcomings of using explicit feature map copy for splicing can be alleviated; (2) balance the computation of each layer—generally, the number of channels at the base layer of DenseNet is much greater than the growth rate, and since the number of channels at the base layer involved in dense layer operations in some partial dense blocks only accounts for half of the original number, nearly half of the computational bottleneck can be effectively removed; and (3) reduce the memory flow—assuming that the size of the basic feature map of a dense block in the DenseNet is $w * h * c$, the growth rate is $d$, and there are $m$ dense layers in total. Then, the CIO of the dense module is calculated as follows:

$$
(c * m) + \frac{(m^2 + m) * d}{2}
\tag{3}
$$

The CIO of partial dense blocks is calculated as follows:

$$
\frac{(c * m) + (m^2 + m) * d}{2}
\tag{4}
$$

Although $m$ and $d$ are usually much smaller than $c$, the a partial dense module can still save up to half of the memory flow in the network.

### 2.3.3. Partial Transition Layer

The purpose of designing a partial transition layer is to maximize the difference of gradient combinations. The partial transition layer is a hierarchical feature fusion mechanism that adopts the strategy of truncating the gradient flow to prevent different layers from learning repeated gradient information. Here, two variants of CSPDenseNet are used to show how this truncation affects the learning ability of the network. Figure 5a shows four different fusion strategies. CSP (fusion first) refers to connecting the feature maps generated by two parts, and then performing conversion. If this strategy is adopted, a lot of gradient information will be reused. As for the CSP (fusion last) strategy, the

output of dense blocks will pass through the transition layer, and then be connected to the feature map from the first part. If the CSP (fusion last) strategy is adopted, gradient information will not be reused, because the gradient flow is truncated. The results shown in Figure 5a are for the case where the four architectures shown in Figure 5b are used for image detection. If the CSP (fusion last) strategy is used for image detection, the computational cost will be significantly reduced, but the accuracy of the first part is only reduced by 0.3%. On the other hand, the CSP (fusion first) strategy does significantly reduce the computational cost, but the highest accuracy is significantly reduced by 1.9%. In the process of information integration, the cross-stage splitting and merging strategy can effectively reduce the possibility of repetition. It can be seen from the results in Figure 5 that the learning ability of the network will be greatly improved if repeated gradient information is effectively reduced.

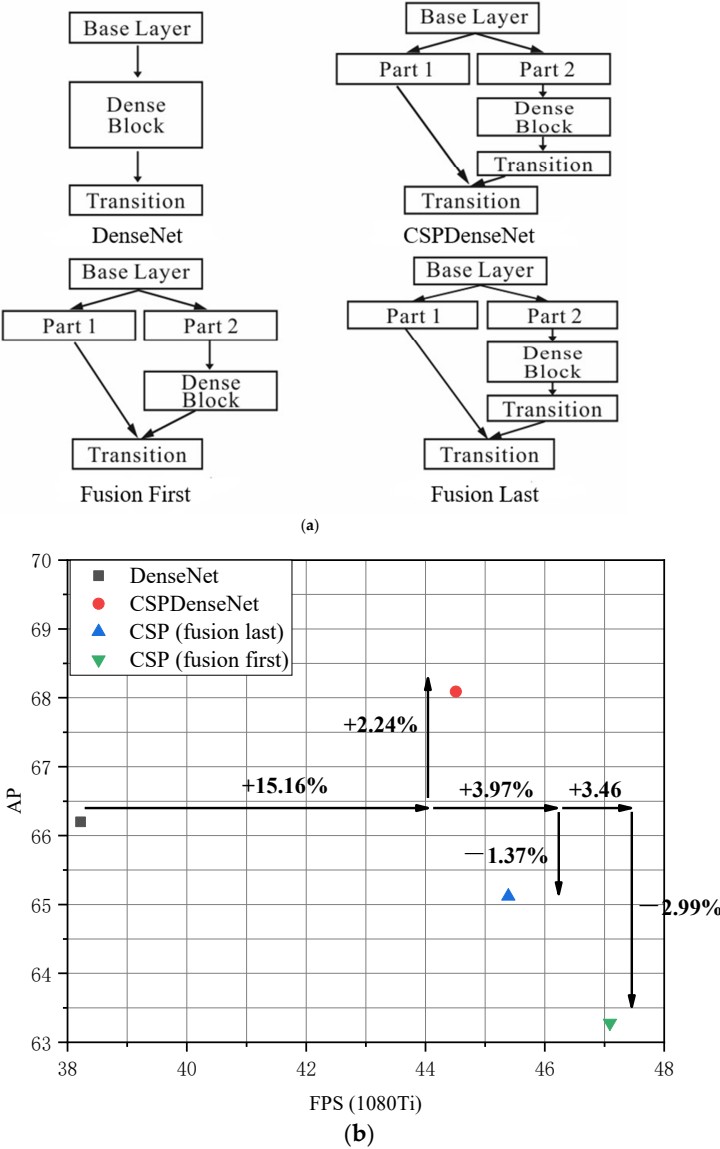

**Figure 5.** Four image detection architectures and results as shown in Figure 5(**a**) were used for image detection, with DenseNet as the baseline, and the detection results were as shown in Figure 5(**b**). Obviously, the CSPDenseNet strategy for image detection reduced the computational cost by 15.16%, and increased AP by 2.24%, while the CSP (Fusion Last) strategy reduced the computational cost by19.13% though decreased AP by 1.37%. Similarly, the CSP (Fusion First) reduced the computational cost by 22.59% though de-creased AP by 2.99%.

2.3.4. CBL + SPP

CBL is the smallest component in the whole network structure, which is closely connected by one convolutional layer, batch normalization (BN), and leaky ReLU activation functions. In order to maximize the difference in gradient flow passing through CBL, an SPP component is added to the top layer of the CBL component, and kernel sizes and strides of varied sizes are used to output different receptive field features; then, a concat operation is performed. Specifically, the equations of the feature mapping matrix of the CBL + SPP component are as follows:

$$
\begin{aligned}
K_h &= ceil\left(\frac{h_{in}}{n}\right) \\
S_h &= ceil\left(\frac{h_{in}}{n}\right) \\
P_h &= floor\left(\frac{k_h * n - h_{in} + 1}{2}\right) \\
H &= 2 * P_h + h_{in}
\end{aligned}
\tag{5}
$$

$$
\begin{aligned}
K_w &= ceil\left(\frac{w_{in}}{n}\right) \\
S_w &= ceil\left(\frac{w_{in}}{n}\right) \\
P_w &= floor\left(\frac{k_w * n - w_{in} + 1}{2}\right) \\
w &= 2 * P_w + w
\end{aligned}
\tag{6}
$$

where $K_h$, $S_h$, $P_h$, and $h$ are the height of the kernel, the step size in the height direction of the feature mapping matrix, the number of fillings in the height direction of the feature mapping matrix, and the height of the feature mapping matrix, respectively; *ceil*() is the rounding up symbol; *floor*() is the rounding down symbol; and $h_{in}$ is the height of the input data.

By combining Equations (5) and (6), the equation of the feature mapping matrix is obtained, as follows:

$$
\left[\frac{h + 2p - f}{s} + 1\right] * \left[\frac{w + 2p - f}{s} + 1\right]
\tag{7}
$$

where $p$ stands for padding, $s$ stands for stride, and $f$ is the input data size.

*2.4. Neck*

The neck component uses the same structure as the backbone. It should be noted that the adopted CSPResNeXt structure can greatly promote the extraction of low-level chrysanthemum features and high fusion at different feature scales. However, when redundant gradient flow enters the neck, in order to truncate the redundant gradient flow and avoid excessive GPU calculation and over-fitting, the main network of the neck component is changed from CSPDenseNet to CSPResNeXt. The CSPResNeXt structure is composed of five groups of blocks of different sizes. The number of ResNeXt networks of each group of blocks is designed to be 1. ResNeXt, in essence, is a group convolution, which controls the number of groups by variable cardinalities. Group convolution is a compromise between normal convolution and depth-wise separable convolution—that is, the number of channels of the feature map generated by each branch is $n$ ($n > 1$). The equation is as follows:

$$
y = x + \sum_{i=1}^{C} T_i(x)
\tag{8}
$$

where + is a shortcut; $C$ is the variable cardinality of the group convolution; $T_i$ is a series of convolutions; and $T$ is composed of continuous convolutions (1*1→3*3→1*1).

*2.5. Head*

The Head component is the prediction part of the network, and the scale of the final predicted feature map is $76 \times 76$, $38 \times 38$, and $19 \times 19$. The structure of the prediction part is FPN + PAN. FPN is a prediction fusion network that conveys strong semantic features from the top to the bottom, and PAN is a prediction fusion network that conveys

strong positioning features from the bottom to the top. The parameters of the two different detection layers are aggregated from different backbone layers, and the new fusion structure is shown in Figure 6.

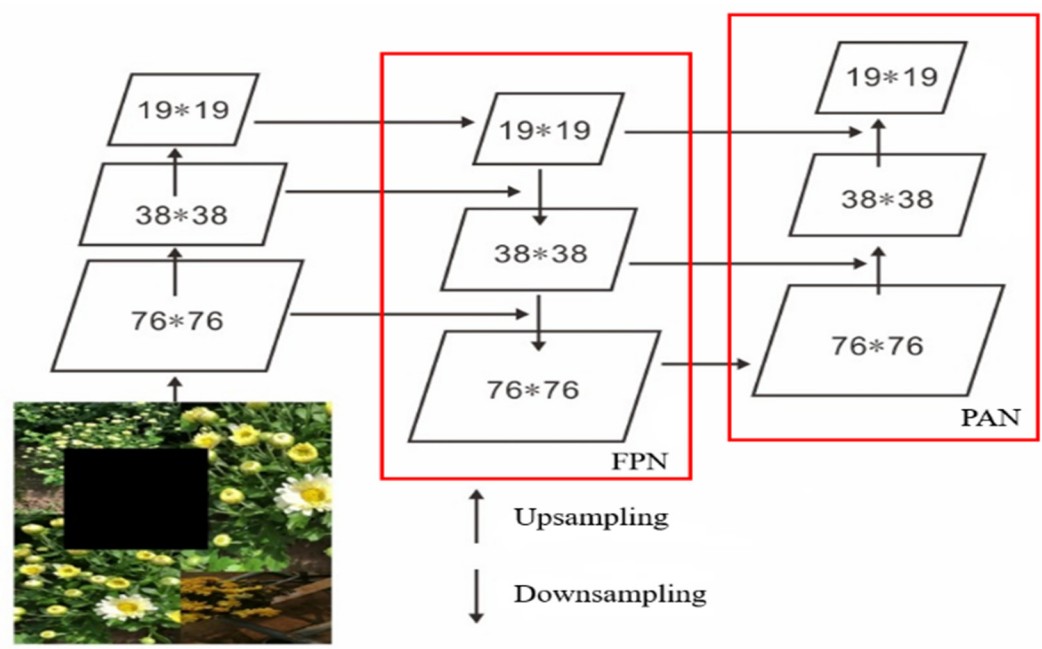

**Figure 6.** FPN + PAN fusion structure. Feature pyramid network (FPN), pyramid attention network (PAN).

## 3. Results

In this section, we designed three experiments to analyze the performance of the proposed detection model. The first experiment was the ablation experiment, which was designed to analyze the contribution of input components and feature fusion components. Different configurations were used to examine the performance of the proposed algorithm. In the second experiment, in order to verify the impact of the dataset size on chrysanthemum detection task, we randomly selected and established eight datasets of varied sizes for comparison. In the third experiment, the robustness of the proposed model under different unstructured environments was studied and tested. Moreover, to describe the experimental process more clearly, the experimental setup and dataset are introduced at the beginning of this section.

### 3.1. Ablation Experiments

To analyze the contribution of input components, we used different configurations to examine the performance of the proposed algorithm. It is worth noting that among the components used for simulating image occlusion, cutout has the best performance, and among the components used for simulating image mosaicking, mosaic has the best performance. The blur component, which is used for simulating image blurring, plays a minor role in improving network performance. Clearly, configuration combinations of different input components can significantly improve the model's performance, but too many configuration combinations may lead a to significant decline in the model's performance. This is because a large amount of repeated gradient information can significantly reduce the learning ability of the network. When the combination of cutout, mixup, cutmix, mosaic, and blur was used, performance reached 68.22%, 88.26%, 70.66%, 49.23%, 72.67%, and 83.18%. When mixup and cutmix were removed from the combination, the AP, $AP_{50}$, $AP_{75}$, $AP_S$, $AP_M$, and $AP_L$ increased by 0.46%, 0.57%, 0.27%, 0.4%, 0.21%, and 0.41%, respectively. When mixup, cutmix, and blur were removed from the combination, the model achieved the best performance, and the AP, $AP_{50}$, $AP_{75}$, $AP_S$, $AP_M$, and $AP_L$ were improved by

0.65%, 1.27%, 0.58%, 1.39%, 0.55%, and 1.45%, respectively. The results of the ablation experiment are shown in Table 2.

**Table 2.** The ablation experiment results of input module performance. Average Precision of small object (APs).

| Random Erase | Cutout | Grid Mask | Mixup | Cutmix | Mosaic | Blur | AP | $AP_{50}$ | $AP_{75}$ | $AP_S$ | $AP_M$ | $AP_L$ |
|---|---|---|---|---|---|---|---|---|---|---|---|---|
| ✓ | | | | | | | 59.22 | 80.54 | 62.34 | 43.56 | 62.33 | 74.83 |
| | ✓ | | | | | | 61.11 | 81.66 | 63.46 | 43.99 | 63.39 | 75.33 |
| | | ✓ | | | | | 59.99 | 81.29 | 62.99 | 43.58 | 63.19 | 75.24 |
| | | | ✓ | | | | 62.59 | 81.92 | 63.89 | 44.64 | 64.12 | 76.67 |
| | ✓ | | ✓ | | | | 63.54 | 82.83 | 64.54 | 45.52 | 64.67 | 77.28 |
| | | ✓ | ✓ | | | | 62.98 | 83.12 | 65.56 | 45.89 | 65.32 | 77.89 |
| ✓ | | | | ✓ | | | 63.46 | 82.54 | 64.29 | 45.45 | 64.59 | 77.11 |
| | ✓ | | | ✓ | | | 64.06 | 83.89 | 65.87 | 46.28 | 65.88 | 78.36 |
| | | ✓ | | ✓ | | | 63.65 | 82.66 | 64.82 | 45.97 | 64.83 | 77.53 |
| | ✓ | | ✓ | ✓ | | | 64.23 | 84.09 | 66.45 | 47.26 | 69.34 | 78.99 |
| | ✓ | | ✓ | ✓ | ✓ | | 67.63 | 87.52 | 69.83 | 49.11 | 70.26 | 82.24 |
| | ✓ | | ✓ | ✓ | ✓ | ✓ | 68.22 | 88.26 | 70.66 | 49.23 | 72.67 | 83.18 |
| | ✓ | | | | ✓ | ✓ | 68.68 | 88.83 | 70.93 | 49.63 | 72.88 | 83.59 |
| | ✓ | | | | ✓ | | 68.87 | 89.53 | 71.24 | 50.26 | 73.22 | 84.63 |

To analyze the contribution of feature fusion components, we used different configurations to examine the performance of the proposed algorithm. When using the F-YOLO (authors') network as the baseline, the AP, $AP_{50}$, $AP_{75}$, $AP_S$, $AP_M$, and $AP_L$ values were 68.87%, 89.53%, 71.24%, 50.62%, 73.22%, and 84.63%, respectively. When the CSPResNeXt, SPP, and PAN feature fusion modules were removed separately, the AP decreased by 2.5%, 0.61%, and 0.35%, respectively. It can be seen that CSPResNeXt provides the best fusion effect for the F-YOLO network, and it was verified that replacing the main network CSPDenseNet with CSPResNeXt in the neck component can significantly improve the learning ability of the cross-stage network. When the CSPResNeXt, SPP, and PAN feature fusion modules were removed at the same time, the $AP_S$, $AP_M$, and $AP_L$ values decreased by 2.75%, 2.00% and 2.17%, respectively. This shows that the feature fusion modules in this paper can achieve the best results for small object detection. It is well-known that improving the performance in small object detection is very challenging. The good news is that the detection speed of the whole model increased from 32.59 to 44.36 FPS after the model was regularly equipped with three feature fusion modules, and the detection speed, which plays a key role in the later deployment of the network on the mobile GPU, was increased by 32.12%. In addition, when CSPResNeXt was replaced by the original main network CSPDenseNet, the detection speed FPS and model accuracy AP were reduced by varying degrees, especially for the detection of small objects, and the $AP_S$ was significantly reduced by 3.1%, which proves the excellence of the design in Section 3.2. The data of the ablation experiment are shown in Table 3.

**Table 3.** The results of the ablation experiment of feature fusion components. Frames Per Second (FPS).

| Method | FPS | AP | $AP_{50}$ | $AP_{75}$ | $AP_S$ | $AP_M$ | $AP_L$ |
|---|---|---|---|---|---|---|---|
| Ours-CSPResNeXt-SPP-PAN | 32.59 | 65.39 | 86.99 | 68.37 | 47.87 | 71.22 | 82.46 |
| Ours-CSPResNeXt-SPP | 36.86 | 65.86 | 87.28 | 68.92 | 48.11 | 71.45 | 82.52 |
| Ours-CSPResNeXt-PAN | 36.39 | 66.12 | 87.46 | 68.96 | 48.23 | 71.59 | 82.67 |
| Ours-CSPResNeXt | 42.89 | 66.37 | 87.82 | 69.23 | 48.29 | 71.82 | 82.87 |
| Ours-SPP-PAN | 34.52 | 67.99 | 88.96 | 70.33 | 49.99 | 72.67 | 83.99 |
| Ours-SPP | 39.62 | 68.26 | 89.11 | 70.89 | 50.28 | 72.89 | 84.24 |
| Ours-PAN | 38.23 | 68.52 | 89.23 | 70.98 | 50.43 | 73.12 | 84.54 |
| Ours (CSPDenseNet) | 41.39 | 67.22 | 88.82 | 69.46 | 47.52 | 71.23 | 83.99 |
| Ours | 44.36 | 68.87 | 89.53 | 71.24 | 50.62 | 73.22 | 84.63 |

### 3.2. Impact of Dataset Size on the Detection Task

To verify the impact of the dataset size on the chrysanthemum detection task, in this paper, eight datasets of varied sizes were randomly extracted and established from the chrysanthemum training set, which contained 100, 200, 400, 800, 1600, 3200, 6400, and 12,040 images. The AP, $AP_{50}$, $AP_{75}$, $AP_S$, $AP_M$, and $AP_L$ are shown in Figure 7.

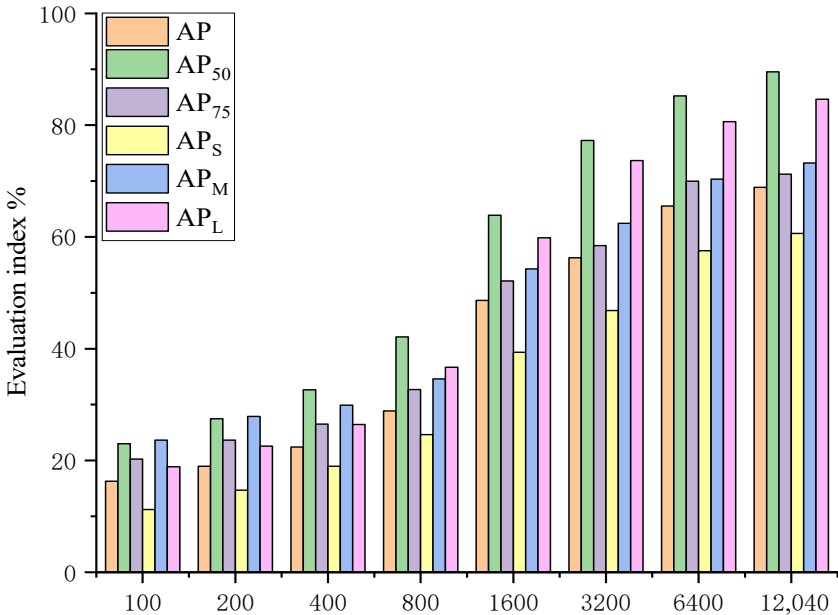

**Figure 7.** Impact of dataset size on the detection task. Average Precision (AP), Average Precision at Intersection over Union = 0.5 ($AP_{50}$), Average Precision at Intersection over Union = 0.75 ($AP_{75}$), Average Precision of small object ($AP_S$), Average Precision of medium object ($AP_M$), Average Precision of large object ($AP_L$).

It can be seen from the table that the performance of the detection model improves with the increase in dataset size. When the number of images was less than 1600, the AP, $AP_{50}$, $AP_{75}$, $AP_S$, $AP_M$, and $AP_L$ increased rapidly with the increase in the number of images. The AP value increased significantly from 16.28% to 48.63%, and the performance of the detection model was improved by 198.71%. When the size of the dataset exceeded 1600 images, the performance improvement rate gradually slowed down and became saturated, and the AP value increased from 48.63% to 68.87%. It is worth mentioning that the dataset size is the most important factor contributing to the improvement in the detection result of large objects. The AP value increased from 18.86% to 84.63%, an increase of 348.73%.

### 3.3. Impact of Different Unstructured Scenarios on the Detection Task

This study examined the robustness of the proposed model in different unstructured environments, including strong light, weak light, normal light, high overlap, moderate overlap, normal overlap, high occlusion, moderate occlusion, and normal occlusion. There were 101,136 chrysanthemums in the early flowering stage in nine unstructured environments, and their states in different environments were not independent of one another. For example, chrysanthemums in images taken under weak light may moderately overlap with each other. We also visualized the features learned by the F-YOLO network. Although it was very difficult to clarify the mechanisms of deep neural networks, the CNN captured some distinguishing features. Naturally, some filters learned edge information from different directions, while others showed some color features, e.g., yellow, green, blue, etc. To demonstrate the effectiveness of the model for chrysanthemum detection, Figure 8 provides partial feature maps obtained with different convolutional layers. Yellow and green indicate a high activation unit, while blue indicates a low activation unit. These

features represented the characteristics in the initial flowering period of chrysanthemum under different non-structured environments in the detection task.

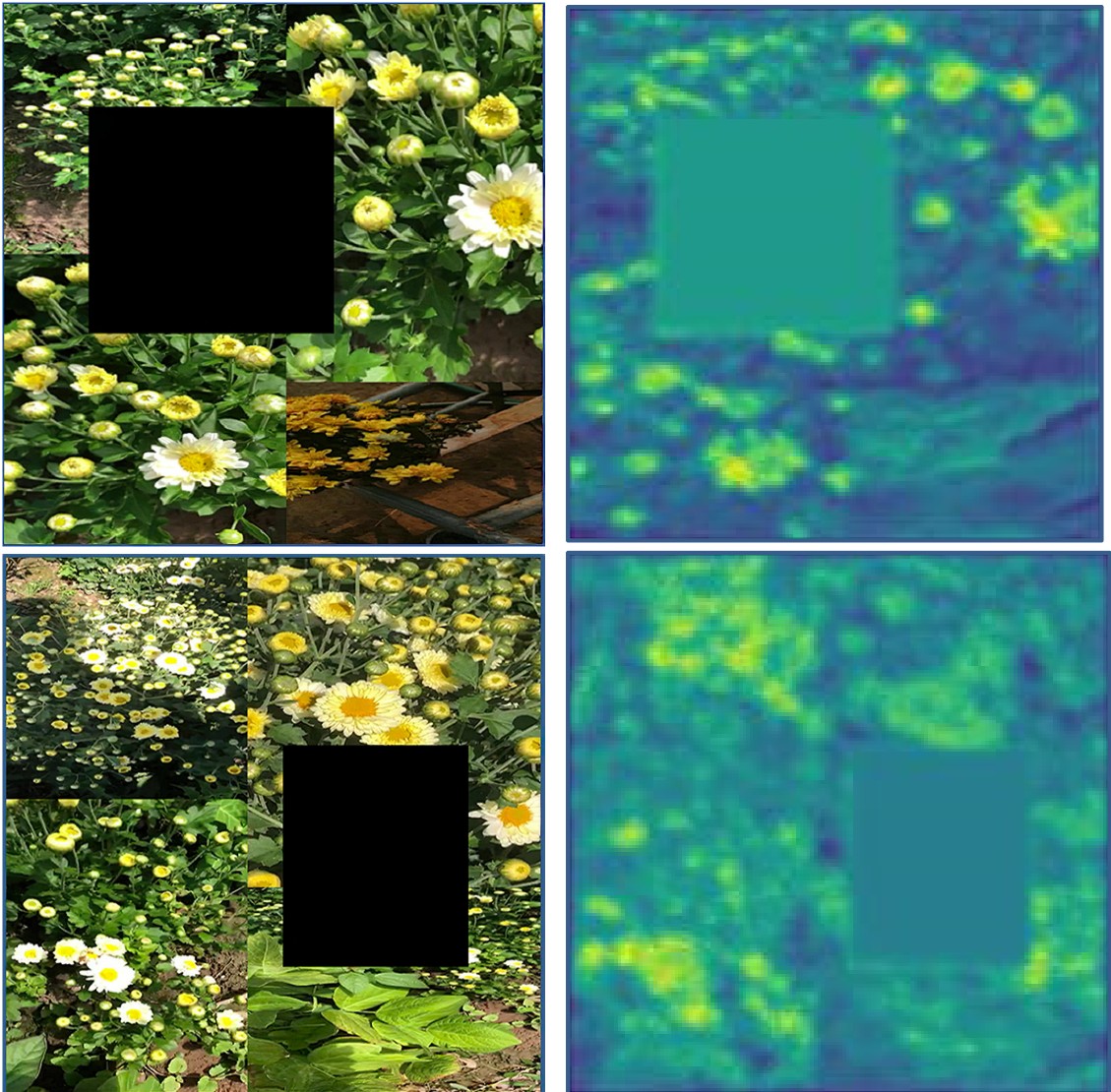

**Figure 8.** Feature maps activated from the convolution layer of the F-YOLO architecture.

Table 4 shows that, under normal environmental conditions, the accuracy of detecting chrysanthemums at the early flowering stage reached impressive values of 98.26%, 98.62%, and 96.29%. It is worth noting that when the unstructured environment became complex, the accuracy of detecting chrysanthemums at the early flowering stage decreased significantly, especially under the scenario of high overlap, where the accuracy was only 86.36%. Interestingly, in all unstructured environments, the error rate under strong light was the highest, reaching 5.26%, while the miss rate under high overlap was the highest, reaching 10.68%. In addition, in general, illumination had the least influence on the detection of chrysanthemums at the early flowering stage. Under strong light, the accuracy, error rate, and miss rate were 88.52%, 5.26%, and 6.22%, respectively. Overlapping had the biggest impact on the detection of chrysanthemums at the early flowering stage. With overlapping, the accuracy, error rate and miss rate were 86.36%, 2.96%, and 10.68%, respectively.

**Table 4.** Impact of different unstructured scenarios on the detection task.

| Environment | Count | Correctly Identified | | Falsely Identified | | Missed | |
|---|---|---|---|---|---|---|---|
| | | Amount | Rate (%) | Amount | Rate (%) | Amount | Rate (%) |
| Strong light | 14,669 | 12,985 | 88.52 | 772 | 5.26 | 912 | 6.22 |
| Weak light | 9941 | 9267 | 93.22 | 439 | 4.42 | 235 | 2.36 |
| Normal light | 76,526 | 75,194 | 98.26 | 574 | 0.75 | 758 | 0.99 |
| High overlap | 11,178 | 9653 | 86.36 | 331 | 2.96 | 1194 | 10.68 |
| Moderate overlap | 36,762 | 35,031 | 95.29 | 529 | 1.44 | 1202 | 3.27 |
| Normal overlap | 53,196 | 52,462 | 98.62 | 314 | 0.56 | 420 | 0.82 |
| High occlusion | 19,957 | 17,608 | 88.23 | 589 | 2.95 | 1760 | 8.82 |
| Moderate occlusion | 54,633 | 51,841 | 94.89 | 322 | 0.59 | 2470 | 4.52 |
| Normal occlusion | 26,546 | 25,561 | 96.29 | 276 | 1.04 | 709 | 2.67 |

*3.4. Comparisons with State-of-the-Art Detection Methods*

To comprehensively verify the performance of the proposed method, we compared the proposed method with other state-of-the art detection methods on the chrysanthemum dataset. The evaluation indexes introduced included FPS, AP, $AP_{50}$, $AP_{75}$, $AP_S$, $AP_M$, and $AP_L$, where the subscripts 50 and 75 denote the threshold of intersection over union (IOU). In addition, subject to the predicted anchor frame size in the experiments, chrysanthemums with an area less than 1394 (34 × 41) were defined as small-size (S) chrysanthemums, those with an area greater than 2888 (76 × 38) were defined as large-size (L) chrysanthemums, and those with an area greater than 1394 and less than 2888 were defined as medium-sized (M) chrysanthemums.

We have compared the proposed algorithm with the latest object detection technologies based on feature fusion, including FPN-based RetinaNet models; EfficientDet [40] models, based on EfficientB + BiFPN; multi-level feature pyramid network (MLFPN)-based M2Det [41] models; YOLOv3 models based on DarkNet53 + FPN; SSD-based models, region-based convolutional neural network (RCNN) models; modified sine–cosine algorithm (MSCA)-based VGG16 + PFPNet [42] models; and RefineDet [43] models, based on Advanced RISC Machine (ARM) + object detection module (ODM). The results of the performance comparison are shown in Table 5.

First, these experimental results were analyzed from the perspective of AP. The AP, $AP_{50}$, $AP_{75}$, $AP_S$, $AP_M$, and $AP_L$ of the proposed method achieved the optimal values of 68.87%, 89.53%, 71.24%, 50.62%, 73.22%, and 84.63%, respectively, which were 0.46%, 1.19%, 1.02%, 0.36%, 5.6%, and 4.41% higher than the YOLOv4 model's values, respectively. YOLOv4 is the top-performing regional detection model based on feature fusion. It can clearly be seen that the proposed method has the best performance in terms of detecting medium-sized objects. There are two reasons for this. The first reason is that the proposed method, based on fusion components, and the alternative fusion of CSPDenseNet + CSPResNeXt can better guide the gradient flow of medium-sized chrysanthemum features; the second reason is that in the complex, unstructured chrysanthemum dataset and the activated, medium-sized chrysanthemum features are more likely to appear in the comfortable environment structure. It is worth mentioning that the detection rate of the proposed method was not the highest, as it was 1.98 FPS slower than that of the YOLOv4 model at the 300 × 300 input scale, but was 155.97% faster than that of the YOLOv4 model at the 608 × 608 input scale. Interestingly, this phenomenon was also seen in other detection models in addition to the YOLOv4 model, such as PFPNet models, YOLOv3 models, SSD models, etc. When small-scale images were used as inputs for these detection models, the detection rate FPS was almost the same as that of the proposed method, or even higher, but the AP was far lower than that of the proposed method. Figure 9 shows the qualitative results of the F-YOLO detection model on the chrysanthemum dataset. As can be observed from the figure, the proposed method can realize real-time, outstanding detection of chrysanthemums at the early flowering stage under complex unstructured scenarios,

including illumination variation, occlusion, and overlapping. The red box represents the chrysanthemums in the initial flowering period. Most of the samples in test set were correctly predicted.

**Table 5.** Comparisons with state-of-the-art detection methods.

| Method | Backbone | Size | FPS | AP | $AP_{50}$ | $AP_{75}$ | $AP_S$ | $AP_M$ | $AP_L$ |
|---|---|---|---|---|---|---|---|---|---|
| RetinaNet | ResNet101 | $800 \times 800$ | 5.92 | 47.33 | 69.89 | 50.11 | 30.23 | 50.39 | 62.22 |
| RetinaNet | ResNet50 | $800 \times 800$ | 6.62 | 54.12 | 76.53 | 56.52 | 35.54 | 57.12 | 68.12 |
| RetinaNet | ResNet101 | $500 \times 500$ | 7.18 | 63.62 | 85.63 | 67.56 | 46.44 | 67.88 | 76.49 |
| RetinaNet | ResNet50 | $500 \times 500$ | 8.03 | 60.89 | 81.11 | 63.06 | 47.34 | 62.96 | 74.33 |
| EfficientDetD6 | EfficientB6 | $1280 \times 1280$ | 6.39 | 70.89 | 88.99 | 70.86 | 51.63 | 72.11 | 78.33 |
| EfficientDetD5 | EfficientB5 | $1280 \times 1280$ | 6.98 | 69.52 | 88.68 | 70.32 | 51.26 | 71.54 | 77.98 |
| EfficientDetD4 | EfficientB4 | $1024 \times 1024$ | 7.36 | 67.22 | 88.23 | 69.45 | 50.66 | 71.23 | 77.84 |
| EfficientDetD3 | EfficientB3 | $896 \times 896$ | 8.22 | 65.33 | 87.39 | 68.84 | 49.06 | 69.05 | 76.83 |
| EfficientDetD2 | EfficientB2 | $768 \times 768$ | 13.99 | 65.06 | 87.33 | 66.31 | 47.49 | 66.32 | 79.98 |
| EfficientDetD1 | EfficientB1 | $640 \times 640$ | 17.11 | 62.45 | 84.56 | 65.91 | 45.38 | 65.99 | 78.33 |
| EfficientDetD0 | EfficientB0 | $512 \times 512$ | 39.89 | 62.53 | 81.34 | 64.31 | 43.11 | 64.54 | 76.56 |
| M2Det | VGG16 | $800 \times 800$ | 6.86 | 59.96 | 81.82 | 62.57 | 40.06 | 62.33 | 75.34 |
| M2Det | ResNet101 | $320 \times 320$ | 15.22 | 54.43 | 77.96 | 56.23 | 39.52 | 56.65 | 68.42 |
| M2Det | VGG16 | $512 \times 512$ | 23.59 | 51.52 | 72.11 | 51.99 | 34.88 | 53.45 | 60.39 |
| M2Det | VGG16 | $300 \times 300$ | 44.19 | 48.33 | 69.06 | 51.42 | 32.52 | 51.42 | 63.46 |
| YOLOv3 | DarkNet53 | $608 \times 608$ | 13.33 | 67.39 | 88.65 | 70.83 | 50.34 | 70.84 | 81.43 |
| YOLOv3(SPP) | DarkNet53 | $608 \times 608$ | 13.46 | 63.92 | 86.36 | 65.87 | 45.63 | 64.56 | 75.54 |
| YOLOv3 | DarkNet53 | $416 \times 416$ | 44.62 | 62.83 | 83.06 | 64.27 | 46.88 | 65.62 | 74.22 |
| YOLOv3 | DarkNet53 | $320 \times 320$ | 45.34 | 56.56 | 77.33 | 59.37 | 38.34 | 59.99 | 72.63 |
| D-SSD | ResNet101 | $321 \times 321$ | 8.46 | 53.29 | 77.24 | 54.94 | 37.89 | 56.42 | 69.88 |
| SSD | HarDNet85 | $512 \times 512$ | 15.89 | 57.88 | 78.52 | 59.39 | 39.92 | 60.22 | 73.29 |
| R-SSD | ResNet101 | $512 \times 512$ | 24.02 | 53.44 | 76.64 | 59.76 | 40.02 | 60.11 | 73.44 |
| DP-SSD | ResNet101 | $512 \times 512$ | 25.42 | 51.26 | 76.88 | 54.45 | 35.33 | 54.53 | 67.99 |
| SSD | HarDNet85 | $512 \times 512$ | 28.11 | 53.56 | 75.99 | 57.83 | 36.45 | 58.64 | 73.27 |
| R-SSD | ResNet101 | $544 \times 544$ | 42.18 | 52.89 | 74.12 | 57.22 | 34.22 | 57.23 | 68.59 |
| DP-SSD | ResNet101 | $320 \times 320$ | 44.54 | 53.44 | 74.52 | 54.38 | 35.62 | 53.89 | 63.34 |
| Cascade RCNN | VGG16 | $600 \times 1000$ | 8.82 | 62.99 | 84.12 | 65.06 | 49.88 | 64.12 | 75.92 |
| Faster RCNN | VGG16 | $224 \times 224$ | 11.34 | 52.39 | 73.54 | 56.35 | 37.67 | 55.45 | 62.52 |
| PFPNet (R) | VGG16 | $512 \times 512$ | 19.65 | 49.22 | 72.89 | 52.22 | 34.68 | 53.22 | 64.67 |
| PFPNet (R) | VGG16 | $320 \times 320$ | 31.23 | 56.38 | 77.13 | 58.36 | 39.24 | 58.92 | 70.85 |
| PFPNet (s) | VGG16 | $300 \times 300$ | 40.99 | 57.99 | 77.62 | 58.89 | 38.99 | 59.93 | 64.86 |
| RFBNetE | VGG16 | $512 \times 512$ | 19.83 | 56.96 | 79.92 | 59.62 | 37.25 | 37.86 | 51.22 |
| RFBNet | VGG16 | $512 \times 512$ | 34.52 | 58.65 | 79.26 | 62.22 | 41.26 | 42.23 | 51.49 |
| RFBNet | VGG16 | $512 \times 512$ | 46.89 | 62.44 | 82.56 | 63.87 | 41.69 | 63.94 | 72.88 |
| RefineDet | VGG16 | $512 \times 512$ | 30.52 | 55.26 | 76.23 | 58.45 | 40.19 | 49.44 | 62.23 |
| RefineDet | VGG16 | $448 \times 448$ | 42.23 | 54.87 | 75.59 | 57.66 | 38.09 | 57.89 | 63.99 |
| DSOD | DS/64-192-48 | $320 \times 320$ | 45.52 | 57.83 | 79.46 | 60.29 | 41.56 | 40.33 | 53.29 |
| ScratchDet | RootResNet34 | $320 \times 320$ | 46.33 | 63.65 | 83.22 | 67.02 | 47.98 | 65.54 | 66.44 |
| YOLOv4 | CSPDarknet53 | $608 \times 608$ | 17.33 | 68.41 | 88.34 | 70.22 | 50.26 | 67.62 | 80.22 |
| YOLOv4 | CSPDarknet53 | $512 \times 512$ | 22.39 | 66.35 | 87.52 | 68.64 | 48.53 | 68.66 | 80.27 |
| YOLOv4 | CSPDarknet53 | $300 \times 300$ | 46.34 | 65.53 | 86.88 | 68.99 | 47.29 | 68.82 | 79.57 |
| F-YOLO (Ours) | CSP | $608 \times 608$ | 44.36 | 68.87 | 89.53 | 71.24 | 50.62 | 73.22 | 84.63 |

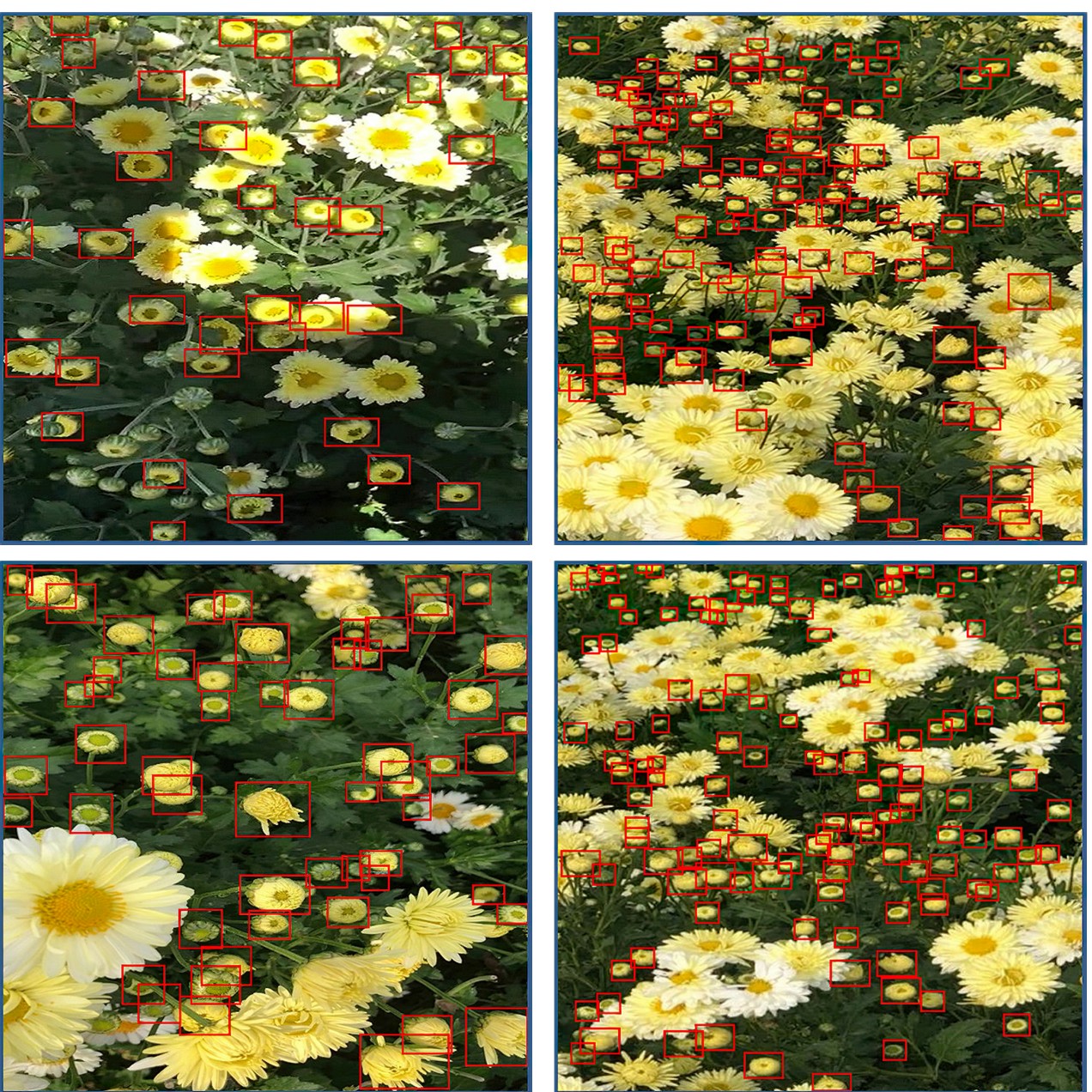

**Figure 9.** Qualitative results of our method.

## 4. Conclusions

In this paper, a lightweight CNN architecture was designed called F-YOLO. The aim was to detect chrysanthemums at the early flowering stage under complex, unstructured scenarios in real time and in an efficient manner. On the one hand, we constructed a separate fusion CNN to fuse different function modules. With these feature fusion modules, the features can be transformed into common subspaces, in which they can be combined linearly or nonlinearly to estimate any complex function. Moreover, to realize the detection task, we fused a variety of data enhancement methods and used specific loss functions to train them. By doing so, these function modules could better understand the chrysanthemum features, thus improving the performance of the lightweight network F-YOLO in complex, unstructured environments in an end-to-end manner. On the other hand, we designed different experiments, including ablation experiments, to examine the performance of the proposed F-YOLO network and evaluate the impact of dataset size

on the chrysanthemum detection task, as well as the robustness of the model in different unstructured environments. Finally, to comprehensively verify the performance of the proposed method, we compared the proposed method with other state-of-the-art detection technologies on the chrysanthemum dataset. The following conclusions can be drawn from this study:

1.  We verified the impact of data enhancement; feature fusion; dataset size; and complex, unstructured scenarios (illumination variation, overlap, and occlusion) on the proposed F-YOLO model. The data enhancement input component equipped with cutout and mosaic achieved the best AP performance of 68.87%. The feature fusion component equipped with CSPResNeXt, SPP, and PAN achieved the best AP performance of 68.87%. The dataset size has a significant effect on the performance of the F-YOLO model, especially for large-sized objects, where the AP increased by 348.73%. In complex, unstructured environments, illumination has the least influence on the detection of chrysanthemums at the early flowering stage. Under strong light, the accuracy, error rate, and miss rate were 88.52%, 5.26%, and 6.22%, respectively. Overlapping has the biggest effect on the detection of chrysanthemums at the early flowering stage. With overlapping, the accuracy, error rate, and miss rate were 86.36%, 2.96%, and 10.68%, respectively;

2.  We compared the performances of the 41 latest object detection technologies, including 12 series model frameworks and the F-YOLO model proposed in this paper. The detection rate FPS of the proposed model was not significantly improved compared with the other latest object detection technologies, but six performance indexes—namely AP, $AP_{50}$, $AP_{75}$, $AP_S$, $AP_M$, and $AP_L$—were significantly improved, and were 0.46%, 1.19%, 1.02%, 0.36%, 5.6%, and 4.41% higher, respectively, than those of the top-performing regional detection model YOLOv4, based on feature fusion in the experiments. In particular, for medium-sized objects, the detection performance $AP_M$ reached 73.22%;

3.  The proposed lightweight model F-YOLO can realize automatic harvesting of chrysanthemums for tea at the early flowering stage, in order to replace manual harvesting. This method can solve the current global situation of relying on manual harvesting of chrysanthemums for tea. Based on the speed and accuracy results from our method, we believe that the advancement of new deep learning architecture and mobile computing devices, together with a large amount of field data, will significantly contribute the development of precision agriculture like chrysanthemum-picking robots in the coming years.

**Author Contributions:** Conceptualization, C.Q.; methodology, C.Q.; software, C.Q.; validation, C.Q., formal analysis, C.Q.; investigation, C.Q.; resources, C.Q.; data curation, C.Q.; writing—original draft preparation, C.Q.; writing—review and editing, I.N.; visualization, I.N.; supervision, K.C.; project administration, K.C.; funding acquisition, K.C. All authors have read and agreed to the published version of the manuscript.

**Funding:** This research was funded by the Northern Jiangsu Science and Technology Major Project—Enriching the People and Strengthening the Power of County Program, grant number SZ-YC2019002.

**Institutional Review Board Statement:** Not applicable.

**Informed Consent Statement:** Not applicable.

**Data Availability Statement:** Data sharing not applicable. No new data were created or analyzed in this study.

**Acknowledgments:** This study is funded by the Northern Jiangsu Science and Technology Major Project—Enriching the People and Strengthening the Power of County Program (grant number "SZ-YC2019002"). I also would like to thank Kun-jie Chen of Nanjing Agricultural University for his technical support.

**Conflicts of Interest:** The authors declare no conflict of interest.

## Abbreviations

| Full Name | Acronym |
| --- | --- |
| Fusion-YOLO | F-YOLO |
| Convolutional neural network | CNN |
| Spatial pyramid pooling | SPP |
| Pyramid pooling module | PPM |
| Atrous spatial pyramid pooling | ASPP |
| Detection with Enriched Semantics | DES |
| Feature pyramid networks | FPN |
| Single Shot multi-box detector | SSD |
| Neural architecture search | NAS |
| Cross-stage partial Densenet | CSPDenseNet |
| Cross-stage partial Resnext | CSPResNeXt |
| Non-maximum suppression | NMS |
| Distance intersection over union | DIOU |
| Stochastic gradient descent | SGD |
| Application-specified integrated circuits | ASICs |
| Perceptual adversarial network | PAN |
| Average precision | AP |
| Parallel feature pyramid network | PFPNet |
| Intersection over union | IOU |
| Multi-level feature pyramid network | MLFPN |
| Modified sine cosine algorithm | MSCA |
| Object detection module | ODM |

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
