# Peer review of "Detecting the Early Flowering Stage of Tea Chrysanthemum Using the F-YOLO Model"

_agronomy, doi:10.3390/agronomy11050834_

Round 1
Reviewer 1 Report
- Please write in passive voice in writing paper, such as 'A fusion detection model was designed ... ' instead of 'We have designed ...' in line 152, 160. Because in a paper what was done is much more important.
- I strongly suggest the authors ask a professional translator to polish the English before submission. There are too many errors of English.
- What is 'AP' in line 210. Please give the full description before using abbreviation. I guess the authors want to write 'Average Precision (AP)'
- 'Where' in line 337 should be 'where' instead of using capital W, because it's inside of a sentence.
- From line 624 'A selective chrysanthemum harvesting robot system will be designed in the future work based on the proposed method for ... ' should be deleted from the conclusion, because the authors only need conclude the experiments results of this paper.
- In line 623, 'the detection performance APM reached 73.22%.' , please verify whether the result is enough for your application. Can the accuracy be improved and how to improve the accuracy, this is important for your research.
Author Response
Response to Reviewer 1 Comments
Thank you very much for giving me the opportunity to revise my manuscript. I will try my best to address each point raised by the reviewers.
Point 1: Please write in passive voice in writing paper, such as 'A fusion detection model was designed ... ' instead of 'We have designed ...' in line 152, 160. Because in a paper what was done is much more important.
Response 1: Thank you for pointing out this mistake. I have read through the text, and have changed the active voice to the passive voice in Lines 155, 163, 172, etc.
Point 2: I strongly suggest the authors ask a professional translator to polish the English before submission. There are too many errors of English.
Response 2: I'm sorry for the inconvenience caused to you by the language problem. I have paid an English Editor recommended by MDPI to improve my English manuscript. The English editing ID provided by MDPI is english-28757. I believe that the quality of my manuscript will be greatly improved with the help of this English editor.
Point 3: What is 'AP' in line 210. Please give the full description before using abbreviation. I guess the authors want to write 'Average Precision (AP)'.
Response 3: I very much admire your understanding of the manuscript. As you pointed out, the AP in Line 215 is Average Precision. I have revised it in the revised manuscript.
Point 4: 'Where' in line 337 should be 'where' instead of using capital W, because it's inside of a sentence.
Response 4: I would like to apologize for the inconvenience caused by the errors. In Lines 264, 361, 369 and 384. I have changed the upper-case W to the lower-case w.
Point 5: From line 624 'A selective chrysanthemum harvesting robot system will be designed in the future work based on the proposed method for ... ' should be deleted from the conclusion, because the authors only need conclude the experiments results of this paper.
Response 5: Thank you very much for your guidance. I agree with you and have deleted the content after Line 663.
Point 6: In line 623, 'the detection performance APM reached 73.22%.' , please verify whether the result is enough for your application. Can the accuracy be improved and how to improve the accuracy, this is important for your research.
Response 6: I admire your sharp eye for detail. In Line 664, APM is the average precision of chrysanthemum images larger than 34 * 41 pixels but less than 76 * 38 pixels. The average precision in this manuscript reaches 73.22%. Here, this indicator is not precision or accuracy, but average precision. In the field of machine vision, average precision over 73.22% is considered acceptable, as this usually means that the precision is more than 85%. For manually
defined medium-sized chrysanthemums, high average precision in the complex unstructured environment proves the feasibility of the model proposed in this manuscript.
Reviewer 2 Report
In this paper, the authors propose a lightweight CNN called Fusion-YOLO, which can adapt to illumination variation, occlusion and overlapping scenarios. The results showed that the Fusion-YOLO model is superior to state-of-the-art technologies in terms of object detection and that this method could be deployed on a single mobile GPU.
The article is well written, and minor details on English grammar require review.
I consider that this article has good potential. However, before being considered ready for publication, a few aspects need to be clarified and improved.
1)First of all, the paper title is too long. An appropriate title should not exceed ten words, be clear, informative and reflect the work.
2)Below the figures(caption part) and tables, there is too much description. The description must be more coherent. The analysis of figures and tables must be inside the text.
3)I kindly suggest the authors add an appendix section and create a glossary with all the acronyms displayed in this work. That will help the reader to understand the content more easily.
4)I recommend the authors include more references from MDPI.
Overall, the novelty was notable and significant.
Author Response
Response to Reviewer 2 Comments
Thank you very much for giving me the opportunity to revise my manuscript. I will try my best to address each point raised by the reviewers.
Point 1: First of all, the paper title is too long. An appropriate title should not exceed ten words, be clear, informative and reflect the work.
Response 1: Thank you very much for your guidance. I have changed the title to “Detecting the Early Flowering Stage of Tea Chrysanthemum Using the F-YOLO Model”, which contains about 10 words.
Point 2: Below the figures(caption part) and tables, there is too much description. The description must be more coherent. The analysis of figures and tables must be inside the text.
Response 2: Thank you for pointing this out. I have added all the descriptive texts that appear below the figures and tables in the manuscript to the main text and try to make the description more coherent in lines 93, 177, 245, 344, 441, 463, 525 and 625.
Point 3: I kindly suggest the authors add an appendix section and create a glossary with all the acronyms displayed in this work. That will help the reader to understand the content more easily.
Response 3: I fully agree with you. In line 688, I have added the glossary as an appendix, and have included all the terms used in the manuscript into the glossary.
Point 4: I recommend the authors include more references from MDPI.
Response 4: Thank you for pointing this out. I have added 6 references from MDPI in addition to the original bibliography.
Reviewer 3 Report
The article is a very pertinent and the Authors have developed an original and interesting topic, full of ideas. Despite this, some integration and clarification are needed: for these reasons I have proposed a "minor revision".
Some comments:
- Section 3.2 “impact of datasheet…”: please consider a slightly broader description.
- Section 4 “Conclusions”: this section is also too concise and schematic: please better focus the conclusions by highlighting the innovative part.
Minor comments:
- Figure 1 is too little and not clear: please improve dimension and resolution.
- Figure 2 is too little and not clear: please improve dimension and resolution.
- Figure 3 is no readable: please improve dimension and resolution.
- Figure 4 is too little and not clear: please improve dimension and resolution.
- Figures 5/a and 5/b are too little and not clear: please improve dimensions and resolutions.
- Figure 6 is no readable: please improve dimension and resolution.
- Figure 8 is too little and not clear: please improve dimension and resolution.
- Figure 9 is too little and not clear: please improve dimension and resolution.
- It is recommended an extensive reading to correct some sentences and typo errors.
Author Response
Response to Reviewer 3 Comments
Thank you very much for giving me the opportunity to revise my manuscript. I will try my best to address each point raised by the reviewers.
Some comments:
Point 1: Section 3.2 “impact of datasheet…”: please consider a slightly broader description.
Response 1: I fully agree with you. As advised, I have added that "The size of data set usually impacts the accuracy of model training to a certain degree" at the beginning of Section 3.2, in order to better express the model performance by adding some broad descriptions.
Point 2: Section 4 “Conclusions”: this section is also too concise and schematic: please better focus the conclusions by highlighting the innovative part.
Response 2: Thank you for pointing this out. As advised, I have added at the beginning of Section 4 that "Based on the speed and accuracy results from our method" in the conclusion part to elaborate the contribution of this technology to the agricultural field by highlighting the innovative part of this technology.
Minor comments:
In order to simulate the operation of chrysanthemum harvester in real life, the data sets in this manuscript were collected by randomly moving video terminals. Due to the complex unstructured environment and mobile video shooting, the resolution of chrysanthemum image samples was low, which was considered one of the technical difficulties in this manuscript. In addition, all the images displayed in the manuscript were tested original images. My intention was to show the test results more intuitively, and as a result, I paid no attention to the acceptance of readers, making it difficult for readers to understand. Due to space limitations, I have deleted some images containing duplicate information, enlarged the original images, and readjusted the layout, in the hope of meeting your requirements.
Point 1: Figure 1 is too little and not clear: please improve dimension and resolution.
Response 1: Thank you for pointing this out. According to your request, I have adjusted the size and resolution of these images.
Point 2: Figure 2 is too little and not clear: please improve dimension and resolution.
Response 2: Thank you for pointing this out. According to your request, I have adjusted the size and resolution of these images.
Point 3: Figure 3 is no readable: please improve dimension and resolution.
Response 3: Thank you for pointing this out. According to your request, I have adjusted the size and resolution of these images.
Point 4: Figure 4 is too little and not clear: please improve dimension and resolution.
Response 4: Thank you for pointing this out. According to your request, I have adjusted the size and resolution of these images.
Point 5: Figures 5/a and 5/b are too little and not clear: please improve dimensions and resolutions.
Response 5: Thank you for pointing this out. According to your request, I have adjusted the size and resolution of these images.
Point 6: Figure 6 is no readable: please improve dimension and resolution.
Response 6: Thank you for pointing this out. According to your request, I have adjusted the size and resolution of these images.
Point 7: Figure 8 is too little and not clear: please improve dimension and resolution.
Response 7: Thank you for pointing this out. According to your request, I have adjusted the size and resolution of these images.
Point 8: Figure 9 is too little and not clear: please improve dimension and resolution.
Response 8: Thank you for pointing this out. According to your request, I have adjusted the size and resolution of these images.
Point 9: It is recommended an extensive reading to correct some sentences and typo errors.
Response 9: I'm sorry for the inconvenience caused to you by the language problem. I have paid an English Editor recommended by MDPI to improve my English manuscript. The English editing ID provided by MDPI is english-28757. I believe that the quality of my manuscript will be greatly improved with the help of this English editor.